# Gold Nanoparticles Inhibit Extravasation of Canine Osteosarcoma Cells in the Ex Ovo Chicken Embryo Chorioallantoic Membrane Model

**DOI:** 10.3390/ijms24129858

**Published:** 2023-06-07

**Authors:** Anna Małek, Marek Wojnicki, Aleksandra Borkowska, Michał Wójcik, Gabriela Ziółek, Roman Lechowski, Katarzyna Zabielska-Koczywąs

**Affiliations:** 1Department of Small Animal Diseases and Clinic, Institute of Veterinary Medicine, Warsaw University of Life Sciences, Nowoursynowska 159c, 02-776 Warsaw, Poland; anna_malek@sggw.edu.pl (A.M.); ziolek.gabriela@gmail.com (G.Z.); roman_lechowski@sggw.edu.pl (R.L.); 2Faculty of Non-Ferrous Metals, AGH University of Science and Technology, Al. A. Mickiewicza 30, 30-059 Kraków, Poland; marekw@agh.edu.pl; 3Faculty of Chemistry, University of Warsaw, Pasteura 1, 02-093 Warsaw, Poland; aleksandraborkow@gmail.com (A.B.); mwojcik@chem.uw.edu.pl (M.W.); 4Faculty of Pharmacy, The Medical University of Warsaw, Banacha 1, 02-097 Warsaw, Poland

**Keywords:** dogs, osteosarcoma, chorioallantoic membrane, extravasation, gold nanoparticles, metastasis

## Abstract

Canine osteosarcoma (OS) is an aggressive bone tumor with high metastatic potential and poor prognosis, mainly due to metastatic disease. Nanomedicine-based agents can be used to improve both primary and metastatic tumor treatment. Recently, gold nanoparticles were shown to inhibit different stages of the metastatic cascade in various human cancers. Here, we assessed the potential inhibitory effect of the glutathione-stabilized gold nanoparticles (Au-GSH NPs) on canine OS cells extravasation, utilizing the ex ovo chick embryo chorioallantoic membrane (CAM) model. The calculation of cells extravasation rates was performed using wide-field fluorescent microscopy. Transmission electron microscopy and Microwave Plasma Atomic Emission Spectroscopy revealed Au-GSH NPs absorption by OS cells. We demonstrated that Au-GSH NPs are non-toxic and significantly inhibit canine OS cells extravasation rates, regardless of their aggressiveness phenotype. The results indicate that Au-GSH NPs can act as a possible anti metastatic agent for OS treatment. Furthermore, the implemented CAM model may be used as a valuable preclinical platform in veterinary medicine, such as testing anti-metastatic agents.

## 1. Introduction

Canine osteosarcoma (OS) is a bone tumor characterized by an aggressive local presentation and high metastatic potential. For dogs, the median survival time varies from 76 to 540 days depending on the method of treatment, which typically involves surgical removal of the primary tumor and adjuvant therapy with chemotherapeutics [1,2]. Metastases mainly occur in the lungs, bones, and the liver [3,4] and often develop parallel with the primary tumor or during treatment. It has crucial clinical significance due to the limited surgical approach towards metastases in veterinary medicine, in comparison to human medicine (e.g., lung metastasectomy) [5]. The mean survival time for dogs with appendicular OS was 188 days, whereas it was only 35 days for those with hepatic metastases [6]. Moreover, it is believed that 80–90% of dogs have a micrometastatic disease at the time of diagnosis, and less than 15% of metastases are radiographically detectable [3,7]. There is still no successful therapeutic approach to prevent metastatic occurrence or progression. Thus, identifying potential anti-metastatic compounds to prevent micrometastatic disease and metastatic spread, not only focusing on primary tumor response, is essential [8].

In metastasis research, models that will simultaneously guarantee a proper environment for complex metastatic processes with easy and fast access are needed. Typically, in vitro models, such as the cancer-endothelium adhesion and invasion assay or Transwell assay, have limitations associated with the 2D design of the study, the lack of in vivo hemodynamics and a full range of blood ingredients, and the interactions between them [9,10]. Establishing an accurate in vivo model is essential and for this reason; different models have been exploited, featuring mammals, such as mice or rats, and others, such as zebrafish or chick embryos [8].

The chick embryo chorioallantoic membrane (CAM) model constitutes a link between in vitro and in vivo studies. It is characterized by several advantages such as time efficiency, low cost, and better visualization of cancer cells in comparison to other in vivo models. Most importantly, it follows the “3R principles” (replacement, reduction, and refinement). The CAM model has been used in human medicine to analyze the metastatic process of tumor cells in several malignant neoplasms [9,11,12,13,14,15,16], but it has been utilized mainly in the assessment of tumor growth and angiogenesis for OS. Different OS cells were grafted onto CAM to assess their ability to form tumors, starting with Balke et al. [17], who confirmed consistent growth of MNNG-HOS, U2OS, and SAOS cells. Kunz et al. [18] optimized the CAM assay using eight OS cell lines and validated the model against a rat xenograft model [19]. In veterinary medicine, the ability to form tumors on the CAM was reported for D17 cells [20]. In human medicine, the CAM model has been proposed as a promising platform for testing anti-metastatic drugs [21,22]. In addition, this model is well suited for assessing potentially invasive cancer cells’ behavior in the circulation, such as extravasation, and to study the efficacy of drugs or gene therapy to modulate it [9,23].

Nanomedicine-based strategies can be used for targeting different steps of the metastatic cascade, therefore, inhibiting metastasis. Recently, the first in vitro studies revealed that gold nanoparticles (AuNPs) have a potential anti-metastatic effect on various human cancer cells [24,25]. For melanoma cells, AuNPs (15 nm) were observed to inhibit the migration, causing the reverse of epithelial-mesenchymal transition (EMT) and reducing angiogenesis by regulating metalloproteinase-2 (MMP-2), c-Myc, and zonula occludens-1 [26]. AuNP-conjugated to quercetin also inhibited EMT, angiogenesis, and metastasis in two breast cancer cell lines [27]. Treatment with different shapes of AuNP inhibited ovarian adenocarcinoma HEY A8 cell migration and invasion [28].

Therefore, the main objective of the study was to evaluate the potential inhibitory effect of the 4.3 ± 1.1 nm colloidal glutathione-stabilized AuNPs (Au-GSH NPs) on the extravasation of canine OS cells with different aggressiveness phenotypes. For this, we utilized the ex ovo CAM model and determined its applicability as a platform for screening potential anti-metastatic agents in veterinary medicine.

## 2. Results

### 2.1. AuNPs Are Not Cytotoxic for OS Cells

MTT assay was performed to assess if the Au-GSH NPs had any cytotoxic effect on the OS cells. All of the range of the tested Au-GSH NPs concentrations (from 10 to 750 μg/mL) were non-toxic, as cell viability was >80% (Figure 1A,B). Both OS cell lines (OSCA-8 and OSCA-32) treated with 500 μg/mL Au-GSH NPs (the highest concentration that does not cause Au-GSH NPs aggregation) in co-culture for 24 h under the contrast-phase microscope did not show any additional signs of altered morphology involved with potential apoptosis or cell death (Figure 1C,D,F,G) than cells from the untreated group (Figure 1E,H). Trypan Blue staining of harvested cells after the 24 h co-culture with Au-GSH NPs (500 μg/mL) revealed that the cell mortality never exceeded 20% in any of the utilized cell lines.

### 2.2. AuNPs Accumulation in OS Cells

Au-GSH NPs accumulation in treated cells, under the contrast-phase microscope, is shown in Figure 1D,G (marked with red arrows). The intracellular uptake of Au-GSH NPs was further confirmed by transmission electron microscopy (TEM) (Figure 2). In the TEM images, due to the high density of gold, Au-GSH NPs are visible as dark spots (Figure 2B). Subsequent magnifications reveal that they are single nanoparticles located inside the cell.

In order to quantify the absorption of nanoparticles by OS cells, Microwave Plasma Atomic Emission Spectroscopy (MP-AES) was used. The OS cells incubated with AuNPs were separated and rinsed, and then the gold content in the cells was determined. Knowing the gold content and the size of the nanoparticle, the average content of nanoparticles absorbed by the cell was calculated. Unabsorbed AuNPs were effectively removed during rinsing, which was confirmed by the analysis of the filtrate. Obtained results are summarized in Table 1. The calculated percentage of Au-GSH NPs absorbed by the cells was 58.8 ± 21.2% and 22.0 ± 1.4% for OSCA-8 and OSCA-32 cell lines, respectively.

### 2.3. Ex-Ovo CAM Model for OS Cells Extravasation Assessment

We utilized the ex ovo (shell-less) CAM model for quantitative analysis of the extravasation potential of canine OS cells injected intravenously, adapting the ex ovo cultivation conditions from Kim et al.’s protocol [9]. The mean rate of eggs successfully transferred to weighing boats at embryonic development day (EDD) 4 was 85 ± 11%. The biggest concern for shell-less chick embryo culturing was embryo survival, which we were able to keep at 80 ± 13% at EDD7 and 62 ± 15% at EDD11.

Utilized cells were successfully labeled with fluorescent green dye, and the proper density of cells spread in the CAM (Figure 3) was provided by injecting approximately 10^5^ cells, using a micro-injector prepared according to Kim et al.’s protocol [9]. Immediately after the intravenous (iv) injection, cells were first circulating within the CAM vessels’ bloodstream (Appendix A) and eventually got trapped in the capillaries. At t = 0 h, injected cells appeared intravascular within the capillaries labeled with Lectin D649 dye (red) on the same plane of view (Figure 4). The lectin dye, which labels the intraluminal surface of endothelial walls of vessels, helps to reveal the intravascular space (red). After 24 h from cancer cells injection, the viable fluorescently labeled cells assessed as extravasated were counted. When the cells did not appear within the labeled CAM blood vessels and were found on a different plane, migrated into the underlying stroma not labeled by the lectins, they were assessed as extravasated (Figure 4). Canine osteoblasts (CnOB) were used as a negative control. The extravasation rate of CnOB cells was 3.1 ± 0.8%, which was significantly lower than the percentage of extravasated OSCA-8 and OSCA-32 cells. We observed that 17.2 ± 2.4% of OSCA-8 and 16.0 ± 2.8% of OSCA-32 cells were able to extravasate in 24 h time period. There was no statistically significant difference between OSCA-8 and OSCA-32 extravasation efficiency (Figure 5), even though OSCA-8 was characterized by a more aggressive phenotype than OSCA-32. Interestingly, some of the OSCA-8 and OSCA-32 cells at t = 48 h were visualized along bigger blood vessels of the CAM (Figure 6). As the CAM model was successfully utilized to observe the OSCA-8 and OSCA-32 cells’ extravasation efficiencies, we sought to use it further as a platform for testing compounds targeting their metastatic ability, such as Au-GSH NPs.

### 2.4. Inhibition of Cancer Cells Extravasation by Au-GSH NPs

A twenty-four-hour cell co-culture with Au-GSH NPs, prior to iv injection, resulted in the inhibition of extravasation efficiency, statistically significant for OSCA-8 (*p*-value ≤ 0.05) and highly significant for OSCA-32 (*p*-value ≤ 0.01), through the decrease in extravasation rate of Au-GSH NPs-treated cells in comparison to untreated cells (Figure 7). The intracellular accumulation of Au-GSH NPs (marked with red arrows) in cells injected into the CAM, treated before with Au-GSH NPs, was visible at t = 24 h (Figure 8).

### 2.5. Changes in MMP-2 Expression after Au-GSH NPs Treatment

The quantitative simple western analyses with total protein normalization showed that the MMP-2 expression was significantly (*p* ≤ 0.01) lower in the CAM samples injected with the OSCA-8 cells treated with Au-GSH NPs compared to control (without treatment) (Figure 9A and Figure 10A). In contrast, there was a significantly higher MMP-2 expression in the CAM samples injected with treated OSCA-32 cells than in the CAMs injected with untreated cells (Figure 9B and Figure 10B).

## 3. Discussion

Metastasis is considered the main reason for cancer morbidity and mortality. One of the possible approaches to improve the prognosis of highly metastatic canine OS is to focus on inhibiting metastases rather than solely altering primary tumor responses. Assessing the invasiveness of canine OS cells is important for a better understanding of their metastatic patterns. In the presented study, we evaluated the extravasation abilities of canine OS cells (Figure 5) (OSCA-8 and OSCA-32 cell lines, with high and less aggressive phenotypes [29], respectively) using the ex ovo CAM model.

We utilized the CAM model as a link between in vitro and in vivo (rodent) models with respect to 3R guidelines, especially due to the benefit of recreating in vivo hemodynamics in comparison to alternative in vitro methods. Rodent models, in comparison to the CAM model, are more difficult in approach and accessibility, generating higher costs and requiring longer experiment duration [30,31]. In this study, we injected the same number of cells as in several other studies with iv injections to CAM veins [9,12,32]. In total, 1 × 10^5^ cells were enough to see cells evenly dispersed in the CAM (Figure 3) but not aggregating in the vessels of the CAM, which could alter calculations. CnOB cells, used as a negative control, were spread evenly in the capillary bed, similar to OSCA cells (Figure 3) after iv injection. Although, most of the CnOB cells showed no signs of undergoing extravasation, as the cells still appeared intravascular, modeled by CAM vessel shape, on the same plane. The calculated percentage of extravasated CnOB cells after 24 h was only 3.1 ± 0.8%, which was significantly lower than the percentage of extravasated OSCA-8 and OSCA-32 cells. It could be related to the CnOB cells’ lack of aggressive metastatic capacities in invasive cancer cells, for example, driven by activation of the Src-STAT3 pathway or heat shock protein 90 activity [33,34]. It shows the importance of performing a detailed evaluation of every new cell line used to assess the extravasation rate in the CAM model, as performed in our study. Usually, extravasated cells appear more fuzzy, as the light scatters through the hemodynamically active CAM, and change their shape when migrating to the surrounding stroma (Figure 4) [9,35].

Additionally, OSCA-8 and OSCA-32 cells were found migrating in close proximity to bigger blood vessels of the CAM (Figure 6B–D). Previously, Deryugina et al. [15] described a similar effect of cells specifically attracted to blood vessels than just CAM mesoderm for HT-hi/diss cells, defined as vasculotropism. It was also shown that highly invasive human melanoma cells were spreading along or in the immediate proximity of CAM vessels [36]. The calculated percentage of extravasation rate was 16.0 ± 2.8% of OSCA-8 and 17.2 ± 2.4% of OSCA-32 (Figure 5). Although OSCA-8 and OSCA-32 were characterized by different gene profiles consistent with more and less aggressive phenotypes [29,37], there was no statistically significant difference between their extravasation efficiencies. It was previously reported that cell lines from the more aggressive molecular phenotype showed more invasive growth and progression in laboratory animals, which included the ability to metastasize [37]. However, in the current study, we focused on extravasation as one of the steps of the metastatic cascade, which does not exclude potential differences in the other steps. A similar effect of non-statistically significant differences between two variants of human fibrosarcoma cells (high and low metastatic dissemination variants) was observed in the study performed by Deruginya et al. [32]. A study performed on a highly metastatic SW620 colon carcinoma cell line and a non-metastatic SW480 cell variant showed that only a few SW480 cells survived in the CAM after 24–48 h, even though cells from both cell lines were equally rapidly arrested in the CAM vasculature in the beginning and then efficiently extravasated, whereas SW620 cells were capable of proliferating with time [12]. We also observed that OSCA-8 cells started proliferation in their initial extravasation spots (Figure 6E,F) after 48 h, which was not presented by OSCA-32. This observation might have been related to OSCA-8’s greater survival ability.

According to the best knowledge of the authors, this is the first study assessing the extravasation efficiency of canine OS cell lines. In contrast, in human medicine, several studies on both extravasation efficacy and anti-metastatic abilities of different compounds were performed using the CAM model [38,39,40]. For example, Merlos Rodrigo et al. [13] compared neuroblastoma cells growth and metastatic potential together with the treatment with cisplatin and ellipticine using in ovo and ex ovo models of CAM and found a significant (*p* < 0.001) inhibitory effect on extravasation to all investigated organs and distal CAM. For a highly invasive PC-3 prostate cancer cell line, the extravasation efficiency after 24 h was calculated as 50% in a representative set of embryos (N = 4) for each experimental group with the same settings as used in the presented study. This value gradually decreased down to 27% when treated with different doses of saracatinib.

In the current study, Au-GSH NPs, previously shown as a non-toxic drug delivery system [41,42,43,44], significantly inhibited OS cells extravasation abilities by decreasing extravasation rates (Figure 7). As shown in the MTT assay, Au-GSH NPs up to high doses did not induce a cytotoxic effect on treated cells (cell viability > 80%) (Figure 1A,B), which was consistent with other studies performed with metastatic canine OS or feline fibrosarcoma, confirming that Au-GSH NPs alone were non-toxic [42,43,44]. Similar formulation of AuNPs have been also reported to lack cytotoxic effects in many other cells, such as OS, breast cancer, pancreatic cancer cell lines, or human osteoblasts [45]. Glutathione-coated AuNPs (1.2 ± 0.9 nm), evaluated in the murine model, have been found to cause no morbidity at concentrations up to 60 μM [46].

The OSCA-8 and OSCA-32 cells’ extravasation rates in an ex-ovo CAM model was decreased after 24 h co-culture with Au-GSH NPs at 500 μg/mL, which suggested potential anti-metastatic properties of the tested AuNPs. Importantly, Au-GSH NPs did not coagulate in the used concentration. It was also observed that the Au-GSH NPs localized inside of the cells were characterized by a narrow distribution (Figure 2B). This may have indicated selectivity of the cell membrane that blocked the absorption of larger particles. This observation was consistent with literature reports [47]. As previously shown, AuNPs could induce different anti-metastatic effects, which may be related to the AuNPs’ sizes, as they determine the physicochemical properties of the nanoparticles. Our study suggested that 4.3 ± 1.1 nm AuNPs influenced an important step in the metastatic cascade–extravasation. Other studies with similar sizes of AuNPs found that 5 nm and 10 nm AuNPs significantly inhibited the thyroid cancer SW579 cells invasiveness, in contrast to bigger (20–60 nm) nanoparticles [48]. The inhibiting effect on cell migration was observed with 15 nm AuNPs, reversing the EMT of B16F10 melanoma cells. For melanoma tumors established from B16F10 cells, AuNPs increased vascular perfusion and decreased permeability, together with reducing MMP-2 expression levels, which decreased the probability of metastasis [26].

Only a few studies on the role of MMP-2 in cancer progression and metastasis in canine OS were performed. Higher expression of MMP-2 was associated with higher invasiveness of canine OS cell lines and tumor aggressiveness [49,50,51]. Matrix metalloproteinases expression was not only high in cancer cells but could be highly expressed in a tumor microenvironment (TME), which consists of other cells, blood, lymphatic vessels, ECMs, and their complex interactions [52]. MMP-2 was found to promote ECM degradation and metastatic niche formation, which stimulates metastatic cancer cell growth [53]. In human medicine, the higher expression of MMP-2 was confirmed in the CAM incubated with SaOS2 cells in comparison to the CAM without cells when assessed by immunohistochemistry [19]. As a result, we sought to determine the MMP-2 expression alterations in the OS metastasis microenvironment. Simple western analysis revealed higher MMP-2 expression in the TME of the CAM injected with OSCA-32 cells (Figure 9B) in comparison to the CAM injected with OSCA-8 cells (Figure 9A); however, both cell lines showed comparable extravasation rates. These results indicated that the two cell lines may have relied on different mechanisms for extravasation. In addition, the influence of the surrounding microenvironment should be considered. It was possible that the more aggressive OSCA-8 cells were less dependent on the alterations in the metastatic niche than the less aggressive ones, as described by Scott et al. in the case of the metastatic cells driven by cell-intrinsic factors, rather than by variations in the microenvironment [37]. Au-GSH NPs treatment in the two cell lines led to different effects on MMP-2 expression, as it decreased in TME of OSCA-8 cells but not OSCA-32 cells (Figure 9 and Figure 10), which could be related to the Au-GSH NPs’ higher uptake in OSCA-8, in comparison to OSCA-32 cells (Table 1). It shows the complexity of tumor cells and the microenvironment interactions, as in both cell lines Au-GSH NPs decreased extravasation rates. The bi-directional effect of Au-GSH NPs treatment on MMP-2 expression suggested that the mechanism of action of Au-GSH NPs was not solely related to MMP-2 inhibition. Other mechanisms appeared to be implicated in this process, and these mechanisms require further evaluation. Nevertheless, the ex ovo CAM model constituted a suitable platform for assessing the potential involvement of different proteins in cancer cells’ metastatic processes in the TME.

## 4. Materials and Methods

### 4.1. Synthesis of Au-GSH NPs

The complex of glutathione-stabilized gold nanoparticles (Au-GSH NPs) (4.3 ± 1.1 nm) was synthesized and determined using TEM (Figure 11) and dynamic light scattering according to a previously described procedure [41,42]. The zeta potential of the particles was equal to −52.9 ± 10.6 mV [43]. All reagents were purchased from Merck (Darmstadt, Germany) unless otherwise stated. Briefly, tetrachloroauric (III) acid (208 mg) was dissolved in distilled water (78 mL) in a round-bottomed flask using magnetic stirring (300 rpm). Then, reduced L-glutathione (468 mg) was slowly added in portions into the mixture within 15 min. To obtain a uniform consistency and increase the solubility of L-glutathione, a saturated solution of sodium bicarbonate (NaHCO_3_) was added dropwise until the solution turned clear. The excess of NaHCO_3_ needed to be avoided since the nanoparticles would not be precipitated in an alkaline environment. Sodium borohydride (NaBH_4,_ 98.0%) (Alfa Aesar, Haverhill, MA, USA) (200 mg) was dissolved in a small amount of distilled water and then rapidly added into tetrachloroauric acid/L-glutathione mixture under vigorous stirring (1000 rpm). The solution changed color and instantly became dark, which indicated the formation of nanoparticles. The reaction was allowed to react for approximately two hours until there was no gas release observed. Afterwards, cooled below 10 °C methanol (160 mL) was added into the mixture and stirred for 10 min. The precipitates were collected by centrifugation (10,000 rpm, 10 min) and washed with 1 mL MeOH and sonicated. The procedure was repeated three times to remove free glutathione. The obtained nanoparticles were dissolved in distilled water and put into in a membrane of 3500 kDa, and dialysis was performed to remove all remaining traces of impurities for three days. The water was changed twice a day.

### 4.2. Cell Culture

The OSCA-8, OSCA-32 (Kerafast, Boston, MA, USA, kerafast.com, accessed on 28 April 2023) canine OS cell lines, and the CnOB (Merck, Darmstadt, Germany) cell line were used in this study. OSCA-8 was derived from a left-shoulder tumor of a 1-year-old intact male Rottweiler, and OSCA-32 was derived from a tumor in the left wrist of a 9-year-old spayed female Great Pyrenees. Cell line aggressiveness profiles were first characterized based on the original tumor behavior and patients’ survival, and, afterwards, gene expression profiling defined two distinct molecular phenotypes of OS, with OSCA-8 and OSCA-32 classified as more and less aggressive, respectively [29,37]. CnOB was derived from a normal canine bone.

The OSCA-8 and OSCA-32 cell lines were cultivated in Dulbecco’s Modified Eagle Medium (Thermofisher Scientific, Waltham, MA, USA) with the addition of 10 mM HEPES, 10% heat-inactivated fetal bovine serum (FBS), and 100 ug/mL Primocin^®^ (InvivoGen, Toulouse, France). CnOB was cultivated in Canine Osteoblast Growth Medium (Cn417-500) with 100 ug/mL Primocin^®^ (InvivoGen, Toulouse, France).

The cells were maintained under standard conditions (5% of CO_2_, 95% humidity, and 37 °C). The media was changed every 48–72 h, and the cells were split when cell confluence reached 70–80%.

### 4.3. Cell Viability Assay

The MTT assay was performed to determine if the Au-GSH NPs presented any cytotoxic effects. The OSCA-8, OSCA-32 cells were seeded in 96-well plates (Becton Dickinson, Franklin Lakes, NJ, USA) at a concentration of 1.5 × 10^4^ cells per well. After 24 h of incubation, the media containing Au-GSH NPs at various concentrations were added: 10, 20, 50, 100, 200, 500, and 750 μg/mL for the Au-GSH NPs-treated group. Medium without tested substances was added to the control group. The cells were incubated for 24 h. After 24 h, the media were removed, and 0.5 mg/mL of tetrazolium salt was added to each well for 4 h. To complete the solubilization of the formazan crystals, 100 μL of dimethyl sulfoxide was added to each well. Photometric absorbance was measured at 570 nm using Infinite M Nano Tecan (TECAN, Mannedorf, Switzerland). The experiment was repeated in triplicate, with 3 technical measures for each concentration.

### 4.4. Au-GSH NPs Co-Culture Preparation

OSCA-8 and OSCA-32 cells were seeded into 6-well plates (Becton Dickinson, Franklin Lakes, NJ, USA) at a concentration of 5 × 10^5^ per well. After cells reached 70% confluence, the tested compound Au-GSH NPs was introduced at 500 μg/mL concentration (1 mL per well) for 24 h. Then, cells were harvested, washed two times with phosphate-buffered saline, and prepared for further experiments.

### 4.5. Transmission Electron Microscopy

To ensure that Au-GSH NPs location was intracellular, TEM was performed. Cells pellets harvested after 24 h co-culture with Au-GSH NPs were fixed by 2.5% glutaraldehyde and 2% paraformaldehyde solution in 0.1 M phosphate buffer for 12 h in 4 °C. After fixation, cells were rinsed three times for 10 min in 0.1 M phosphate buffer. Next, cells were postfixed in 1% osmium tetroxide for 1 h and stained in 2% uranyl acetate for 40 min, both at room temperature. Dehydration was performed by incubating the samples in increasing acetone concentrations. Finally, cells pellets were embedded in the mixture of acetone and Epon resin, then in pure Epon resin. After two days of polymerization at 60 °C, seventy nanometer-thick sections were cut on an ultramicrotome and collected on TEM copper grids. Electron micrographs were obtained with a Morada camera on a JEM 1400 transmission electron microscope at 80 kV (JEOL Co., Tokyo, Japan).

### 4.6. AuNPs Concentration in OS Cells

In order to determine the number of nanoparticles absorbed by the cells, a series of analyses were performed. First, the cells were mineralized using concentrated nitric acid (65%, Avantor, Gliwice, Poland). Next, concentrated hydrochloric acid was added (36%, Avantor, Gliwice, Poland). As a result of mixing nitric acid with hydrochloric acid in a volume ratio of 1:3, aqua regia was obtained. The transparent yellow solution was obtained. Next, the solution was analyzed by MP-AES (MP-AES Agilent 4200, Tokyo, Japan). An analytic standard solution (SCP Science, LOT: S160615016, 1001 ± 4 µg/mL) was used as a reference material in appropriate dilutions to obtain a 5-point calibration curve. Each calibration point and each measurement were repeated 3 times. Next, the determined concentration was used to calculate the number of nanoparticles per cell, assuming that the density of nano-gold was equal to bulk gold (19.28 g/cm^3^), the average radius of the nanoparticles was 4.3 nm, and the number of cells in the mineralized sample was known.

### 4.7. Ex Ovo Chick Embryo CAM Model

White Ross 308 fertilized chicken eggs were purchased from a local provider (Marylka, Poland) and maintained at 39 °C and 70% humidity. At EDD4, shells were cracked, and embryos were transferred into polystyrene weighing boats (Global Scientific, Northallerton, UK). From that moment, they were incubated without shells in a high humidity environment (containers filled with distilled water). After the experiment’s termination, chick embryos were sacrificed by brief placement at −20 °C and decapitation.

### 4.8. Intravenous Injections of Cancer Cells

At the EDD11, chick embryos were injected with cancer cells pre-labeled with Celltracker green™ (Invitrogen, Waltham, MA, USA) according to the manufacturer protocol. Chick embryos were injected with 100 µL of 1 × 10^6^ cells/mL solution of each one of the utilized cell lines. The injection of cells suspension was only performed when more than 85% of cells were alive (cell suspension was mixed with 0.4% Trypan Blue and cells were counted with a Countess II FL Automatic Cell Counter (Thermo Fisher Scientific, Waltham, MA, USA)). CAM vessels were labeled with an intravenous (iv) injection of 100 µL of Lycopersicon Esculentum (Tomato) Lectin (LEL, TL) DyLight 649 (Thermofisher) at 0.1 mg/mL concentration. Every injection was performed with the microinjector prepared according to Kim et al.’s protocol [9]. During the experiment, chick embryos were maintained at 38–39 °C.

The location of cancer cells was examined after injection in t = 0 h and t = 24 h, distal to the injection site in a region of interest (ROI), marked with silicone ring (Figure 6A) (Zegir, Warsaw, Poland). For each treatment group, N = 10–12 chick embryos at EDD 11 were used for performing the injections and in calculations in further statistical analyses. When assessing the extravasation efficiency of OSCA-8 and OSCA-32 cell lines, every embryo used in the study had 2–4 windows placed, each constituting a single region of interest (ROI) (Figure 6A). Calculations were performed using each ROI as an individual repetition. To calculate the mean extravasation rate, all the windows (ROIs) placed in all embryos in the group were averaged, i.e., the mean extravasation efficiency value showed the value for the whole group injected with treated or untreated cells. This was in agreement with the methodology published by Kim et al. [9]. Basic calculation of extravasation efficiency was performed using the formula:(1)$$\frac{\mathrm{number}\;\mathrm{of}\;\mathrm{extravasated}\;\mathrm{cells}\;\mathrm{at}\;t=24\;h}{\mathrm{number}\;\mathrm{of}\;\mathrm{cells}\;\mathrm{at}\;t=0\;h}\times 100\%=\mathrm{extravasation}\;\mathrm{efficiency}\;(\%)$$

After that time, chick embryos were examined in t = 48 h to assess extravasated cancer cells’ proliferation.

### 4.9. Fluorescent Microscopy Imaging

Fluorescently labeled cancer cells and CAM vessels were visualized in vivo using the wide-field fluorescent microscope Zeiss Axio Examiner Z1 (Zeiss, Oberkochen, Germany) with Zen 2.3 software, with filter sets used: ex. 450/490 nm, AF488, em. 500/550 nm for cells, ex. 625/655, AF660, em. 665/715 for vessels, with 5×, 10× and 40× immersion objective.

### 4.10. Protein Quantification

After quantification of extravasated cancer cells, the CAMs injected with OSCA-8 and OSCA-32 cells, both treated with Au-GSH NPs and untreated, were collected and kept on ice for protein extraction. Samples were homogenized for 1 min on medium speed, using Polytron PT 2500 E Homogeniser (VWR, Gdańsk, Poland) in Protease Inhibitor Cocktail Set with lysis radioimmunoprecipitation assay buffer in concentration 1:100. Following homogenization, the samples were agitated on ice and centrifuged at 16,400× *g* for 15 min at 4 °C. The protein concentrations were measured in the supernatant with BCA Protein Assay Kit with Infinite M Nano Tecan (TECAN, Mannedorf, Switzerland).

### 4.11. Quantitative Protein Expression Analyses

The Jess^TM^ Simple Western system (Protein Simple, San Jose, CA, USA), an automated capillary-based immunoassay method [54], was used to quantify MMP-2 expression. Samples were maintained according to manufacturer’s standard method for 12–230-kDa Jess separation module. Briefly, for the main experiment, samples were mixed with 0.1X Sample buffer and Fluorescent 5X Master mix (Protein Simple, San Jose, CA, USA) to achieve a concentration of 0.1 mg/μL, then denatured at 95 °C for 5 min. Human MMP-2 antibodies were (AF902, Biotechne, Minneapolis, MN, USA) diluted 1:5 in Milk-Free Antibody Diluent (Biotechne, Minneapolis, MN, USA), according to manufacturer protocol, and anti-goat secondary HRP antibodies (043-522, Biotechne, MN, USA) were used. At first, the size-based protein separation (separation time 20 min, separation voltage 375 V) was performed using 24 well plates (one for TME of each OS cells treated or untreated with AuNPs, n = 10 samples per group plus positive and negative controls). The chemiluminescent revelation was established with peroxide/luminol-S (Biotechne, MN, USA). For total protein, Compass 6.0 Simple Western software (Protein Simple, Minneapolis, MN, USA) was used to calculate chemiluminescence intensity. Before performing the main experiment, the antibody saturation (min. 90%) and protein linear range were evaluated to assure the proper concentration of samples and primary antibodies used in the study (Appendix A). Results were visualized as electropherograms, representing a peak of chemiluminescence intensity. The area under the curve (AUC) and the corrected AUC were calculated by the software based on total protein normalization (Appendix A). Only samples with a signal-to-noise ratio (S/N) ≥ 10, a peak height/baseline ratio value ≥ 3 (calculated manually from the data obtained in the software), and ranges in total protein values that did not exceed 20%, according to manufacturer’s guidelines, were used for statistical analyses.

### 4.12. Statistical Analysis

The statistical analysis was conducted using one-way ANOVA and Tukey post hoc test and Student’s *t*-test (for cancer cell extravasation rate and MMP-2 expression before and after Au-GSH NPs treatment, respectively) using GraphPad Prism 8.0 (San Diego, CA, USA). A *p*-value ≤ 0.05 (*) was determined as significant and *p* ≤ 0.01 (**) and *p* ≤ 0.001 (***) as highly significant.

## 5. Conclusions

We showed that Au-GSH NPs significantly inhibited canine OS cell extravasation, regardless of the aggressiveness of the phenotype of the tested cell lines, while not presenting a cytotoxic effect. Therefore, Au-GSH NPs could potentially be used as an anti-metastatic agent for canine OS. By utilizing the CAM ex ovo model for the quantitative analysis of extravasation efficiency, we further demonstrated that the CAM assay was well suited for short-term in vivo studies and may be successfully used in veterinary medicine as a platform for testing potential anti-metastatic drugs and their possible influence on the TME. Interestingly, Au-GSH NPs caused a significant decrease (*p* ≤ 0.05) of MMP-2 expression in the microenvironment of extravasated OSCA-8 cells with a more aggressive phenotype, in contrast to less aggressive ones (OSCA-32). Future studies are necessary to further explore our current results and unravel Au-GSH NPs’ mechanisms.

## Figures and Tables

**Figure 1 ijms-24-09858-f001:**
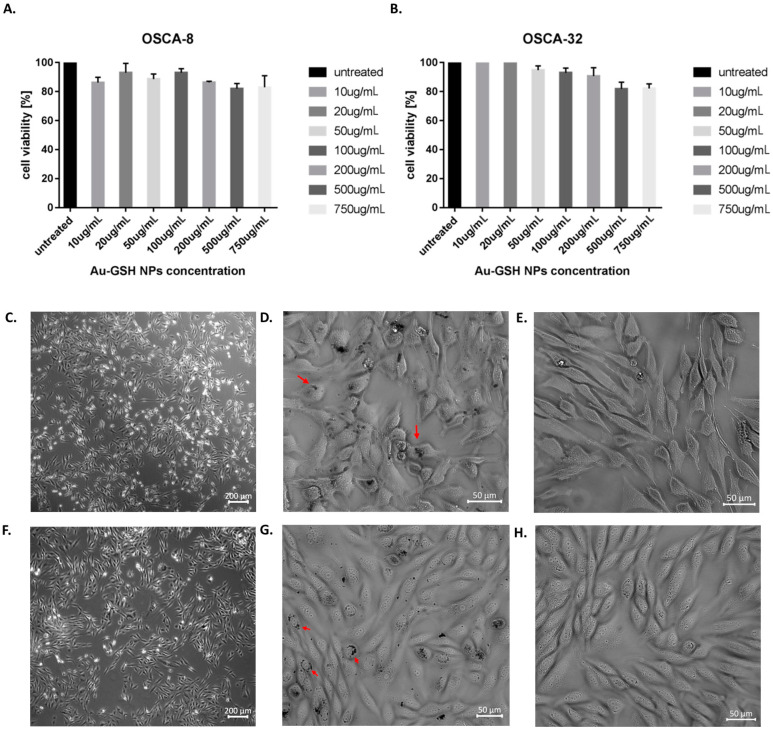
Effect of Au-GSH NPs on OS cell viability in MTT assay (% ± SEM): OSCA-8 (**A**), OSCA-32 (**B**). Phase-contrast microscopy images of treated (**C**,**D**) vs. non-treated (**E**) OSCA-8 cells and treated (**F**,**G**) vs. non-treated (**H**) OSCA-32 cells. Red arrows indicate the intracellular accumulation of Au-GSH NPs (**D**,**G**). 4× magnification, scale bar 200 μm (**C**,**F**). 20× magnification, scale bar 50 μm (**D**,**E**,**G**,**H**).

**Figure 2 ijms-24-09858-f002:**
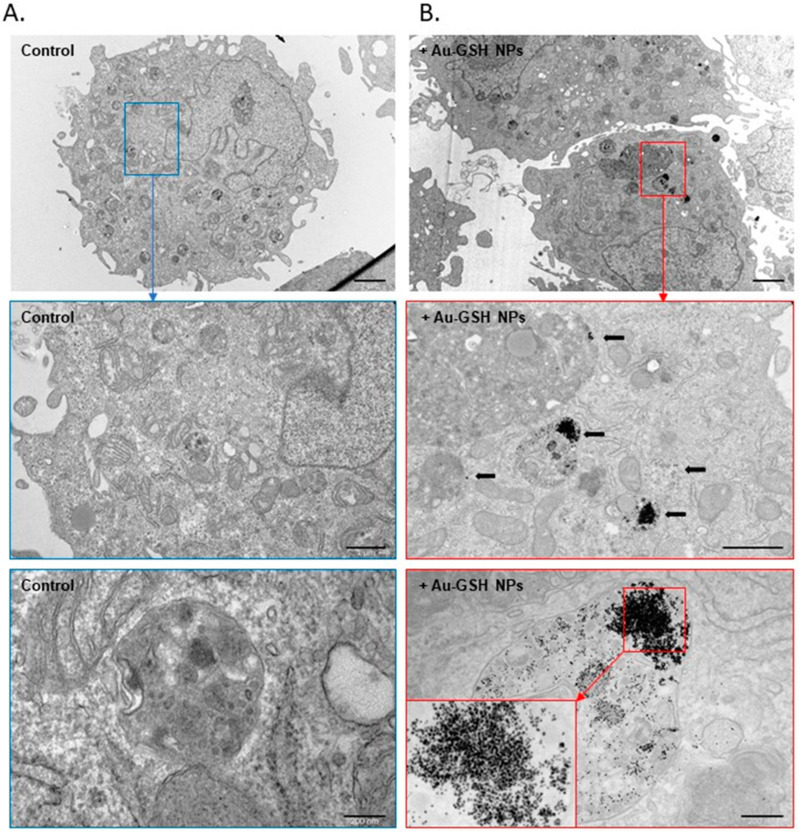
Cellular uptake of Au-GSH NPs by OSCA-8 cells in TEM. Non-treated cells (**A**). Cells treated with Au-GSH NPs for 24 h (**B**). Images are magnified and highlighted with color (blue for non-treated, red for Au-GSH NPs treated) to accentuate the location of nanoparticles within the cell. Black arrows indicate the accumulation of Au-GSH NPs. Scale bar 2 μm, further magnification scale bar 1 μm and 200 nm.

**Figure 3 ijms-24-09858-f003:**
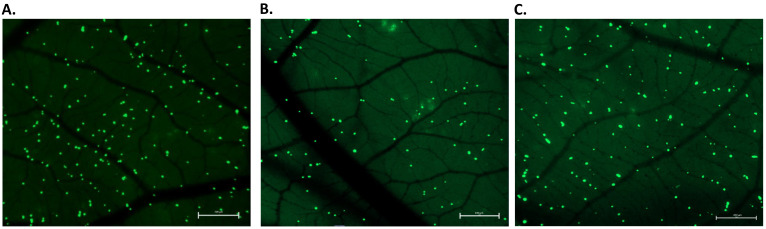
OSCA-8 (**A**), OSCA-32 (**B**), CnOB (**C**), fluorescently labeled with CellTracker™ Green CMFDA, after iv injection into CAM at t = 0 h. Scale bar, 200 µm.

**Figure 4 ijms-24-09858-f004:**
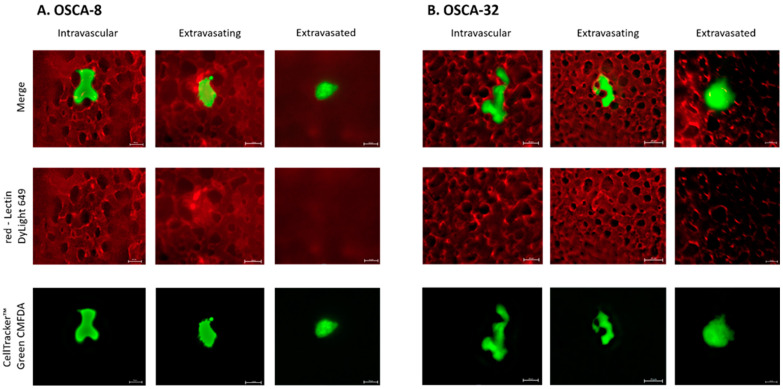
OSCA-8 (**A**) and OSCA-32 (**B**) cells, fluorescently labeled with CellTracker™ Green CMFDA, after iv injection, in the process of extravasation. Red-Lectin DyLight 649 dye was used to reveal blood vessels. Scale bar, 10 µm.

**Figure 5 ijms-24-09858-f005:**
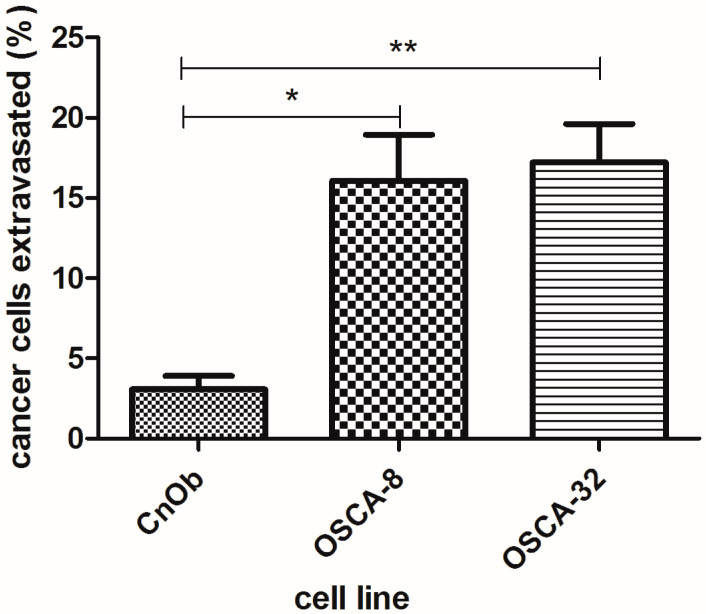
Extravasation efficiency of CnOB, OSCA-8 and OSCA-32 cell lines. Data presented as mean ± SEM): 3.1 ± 0.8% of CnOB, 16.0 ± 2.8% of OSCA-8 and 17.2 ± 2.4% of OSCA-32 cells. *p* ≤ 0.05 (marked as “*”) was assigned as significant and *p* ≤ 0.01 (marked as “**”) as highly significant.

**Figure 6 ijms-24-09858-f006:**
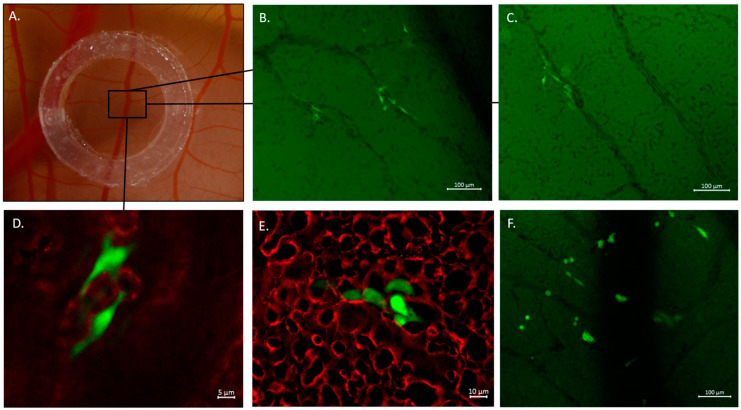
OSCA-8 cells in the evaluated region of interest (ROI) (**A**), migrating in close proximity to CAM blood vessels (**B**,**C**), scale bar 100 µm, visualized along the bigger vessel at t = 48 h (**D**), scale bar 5 µm. OSCA-8 cells proliferating in the initial extravasation spots at t = 48 h (**E**,**F**), scale bar 10 µm (**E**), scale bar 100 µm (**F**).

**Figure 7 ijms-24-09858-f007:**
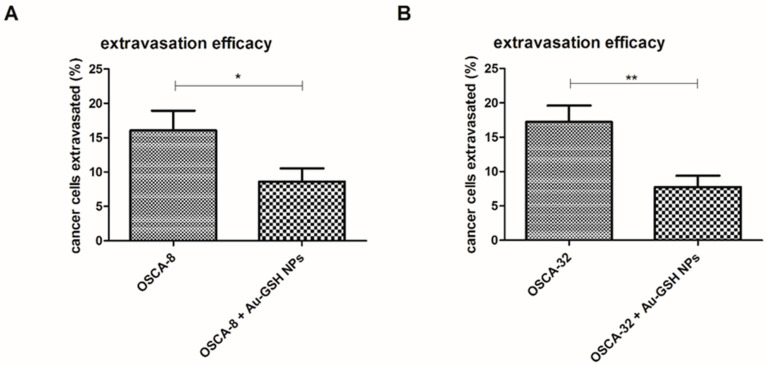
Extravasation efficiency of OS cell lines: OSCA-8 (**A**), OSCA-32 (**B**), with and without Au-GSH NPs treatment, calculated as percentage of extravasated cells. *p* ≤ 0.05 (marked as “*”) was assigned as significant and *p* ≤ 0.01 (marked as “**”) as highly significant. Data presented as mean ± SEM: 8.6 ± 1.9% and 16.0 ± 2.8% for OSCA-8 with and without Au-GSH NPs treatment, respectively, and 7.7 ± 1.6% and 17.2 ± 2.4% for OSCA-32 with and without Au-GSH NPs treatment, respectively.

**Figure 8 ijms-24-09858-f008:**
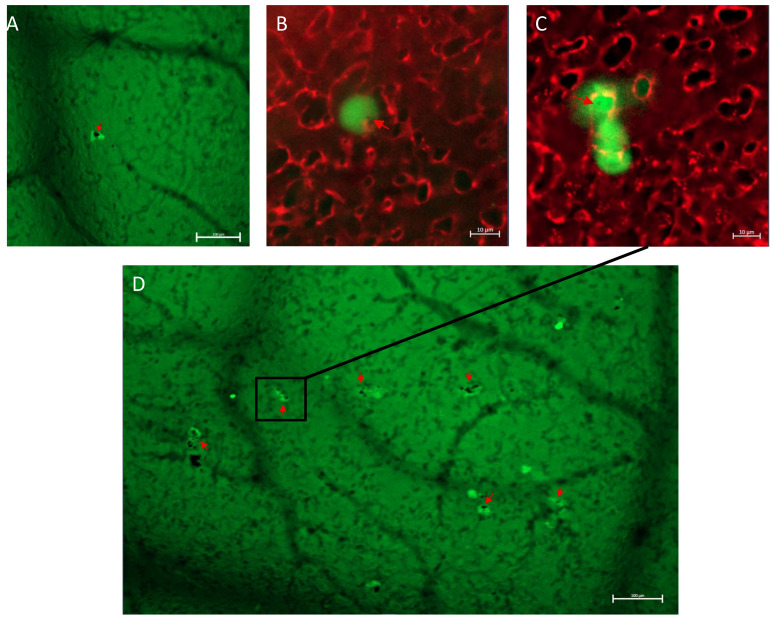
OSCA-32 (**A**,**B**) and OSCA-8 (**C**,**D**) cells in the CAM, with intracellular accumulation of Au-GSH NPs (marked with red arrows). Scale bar 100 μm (**A**,**D**), 10 μm (**B**,**C**).

**Figure 9 ijms-24-09858-f009:**
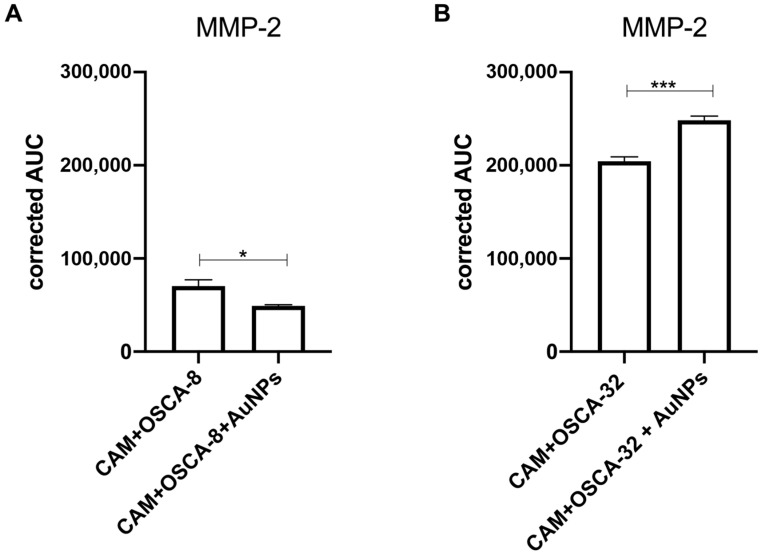
MMP-2 expression in the CAM injected with OSCA-8 (**A**) and OSCA-32 (**B**) cells, untreated or treated with Au-GSH NPs. Data presented as mean ± SEM. *p* ≤ 0.05 (marked as “*”) was assigned as significant, and *p* ≤ 0.001 (marked as “***”) as highly significant.

**Figure 10 ijms-24-09858-f010:**
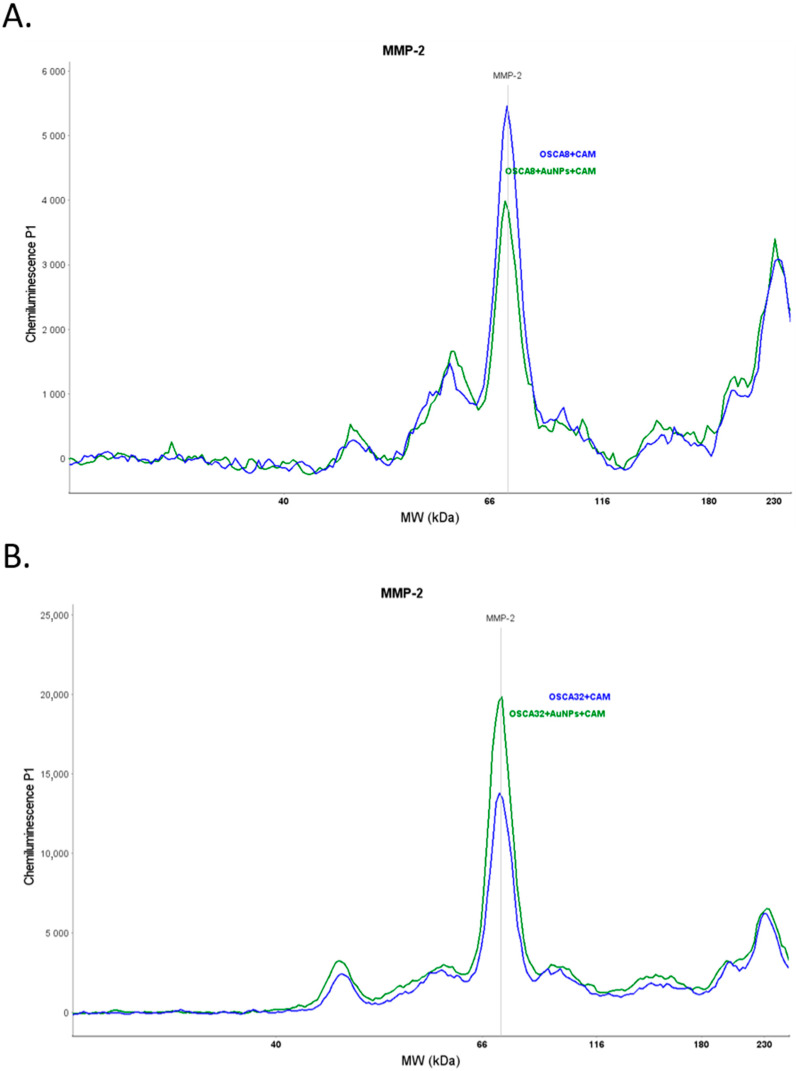
Simple western MMP-2 expression in the example CAM injected with OSCA-8 (**A**) and OSCA-32 (**B**) cells, untreated (blue line) or treated with Au-GSH NPs (green line).

**Figure 11 ijms-24-09858-f011:**
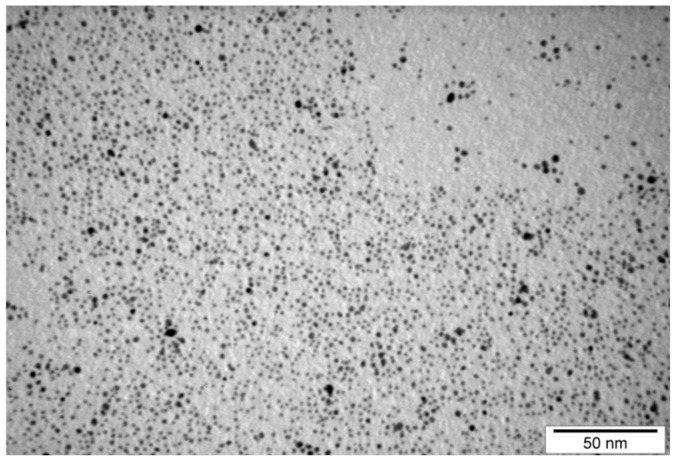
TEM images of glutathione-stabilized gold nanoparticles (Au-GSH NPs).

**Table 1 ijms-24-09858-t001:** Uptake of Au-GSH NPs by OS cells.

Cell Line	Number of Au-GSH NPs/Cell	Au-GSH NPs Absorbed, %
OSCA-8	1.93 × 10^5^ ± 6.05 × 10^4^	58.8 ± 21.2
OSCA-32	9.27 × 10^4^ ± 1.86 × 10^4^	22.0 ± 1.4

## Data Availability

All data generated or analyzed during this study are included in this published article.

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
