# Peer review of "Gold Nanoparticles Inhibit Extravasation of Canine Osteosarcoma Cells in the Ex Ovo Chicken Embryo Chorioallantoic Membrane Model"

_ijms, 2023, doi:10.3390/ijms24129858_

Round 1

Reviewer 1 Report (Previous Reviewer 1)

While authors have reworked the manuscript, there are still areas for improvement

Introduction section currently is too long, each subsection (canine osteosarcoma, metastasis models and nanoparticles) can be shortened to about 50%.

Whole manuscript and specifically Discussion section readability can also be improved – for example there is no need to list every line designation used in referenced study in a paragraph already full of other abbreviations, references and mentions of figures. (lines 320-323)

Some parts of discussion dedicated to topics less relevant to study results (e.g. size of nanoparticles, considering that only one size was used in current study) are too big.

Only short sentence in Results section mentions OSCA-8 line being more aggressive than OSCA-32, authors should add more information about differences between two and in more clear way, and also should discuss their results in that context.

When authors discuss MMP-2 changes in different OSCA lines, there is no mention of lower MMP-2 level in OSCA-8 without AuNPs compared to OSCA-32 while it is characterized as being more aggressive.

Sections in Materials and Methods should be named better- for example instead of TEM non-abbreviated designation should be used, its not clear what BCA analysis is for, use generalized assay name instead of brand Simple Western etc.

Also, lines 531-532 state “Human MMP-2 antibody (AF902, Biotechne, MN, USA)” – company site has only information about human reactivity, authors should clear the question if used antibodies have chicken or canine MMP reactivity.

In Table 1, authors should reformat numbers of NPs/cell to more readable format (eg. 1.9*10^5)

In Figure 6, subsections should be reformatted to same size, also scale bars could be made more prominent.

Quality of Figure 10 should be improved – numbers and text present on figure are almost unreadable.

Authors should recheck manuscript for general language improvement and typos –  some examples:

Figure 6 legend “… (E,F), scale bar 10 μm €,…”

Figure 9 legend “p<0.001 (marked as “**8”)”

Author Response

Reviewer 2 Report (Previous Reviewer 2)

Overall the authors have really improved the quality of their manuscript. Thank you very much.

However, there are still some very minor points that can be changed quickly.

1) Supplementary files

Fig A: chemiluminescence graph: the legends are too small as well as the scales, it is unreadable. Can the authors change the font size, it's a shame not to be able to see them.

Fig B: There are no indications on the x-axis. Can the authors add a scale?

Fig B : primary antibody dilution: As in fig A, the scales and colour legend are not legible. Can the authors change the font size?

Fig C: the scales and colour legend are not legible. Can the authors change the font size?

2) Manuscript

Page 6, table 1: There is no unit indicated for Au-GSH-NPs/cell, can the authors add one?

Page 9: Figure 8 is not mentioned in the text, please add it.

Page 12: Figure 10 is very blurred, can you redraw it so that it is clearer and easier to read? Why don't you put the same scales on the y-axis, this would allow a better comparison of the two cell lines?

Page 13, lines 269 – 272. The sentence is not well formulated; please can the authors rewrite this sentence?

Page 15, line 402: it’s KDa instead kDA, please change it.

Page 17, line 465: it would be really good to indicate the acronym MP-AES (even if there is a table of acronyms used in this manuscript).

Round 2

Reviewer 1 Report (Previous Reviewer 1)

Authors have adressed all my comments

Author Response

Dear Reviewer,

Thank you very much for reviewing one more time the article entitled “Gold Nanoparticles Inhibit Extravasation of Canine Osteosarcoma Cells in the Ex Ovo Chicken Embryo Chorioallantoic Membrane Model” and accepting all our answers for your previous comments.  

Kind regards,

Katarzyna Zabielska-Koczywąs

This manuscript is a resubmission of an earlier submission. The following is a list of the peer review reports and author responses from that submission.

Round 1

Reviewer 1 Report

The authors manuscript “Gold Nanoparticles Inhibit Extravasation of Canine Osteosarcoma Cells in the Ex Ovo Chicken Embryo Chorioallantoic Membrane Model” presents interesting results. Yet, the paper could benefit from further refinement.

Introduction:

Lines 35-37
“For canine OS it is believed that 80-90% of patients have a micrometastatic disease at the time of diagnosis and less than 15% are radiographically 36 detectable”

 – if authors refer to data stating that most of patients already have micrometastases before any treatment starts, they should also elaborate why the ability to prevent metastases would be relevant for clinical use.

Results:

On MTT assay and cells injected in CAM after cultivation – in Figure 1 D and G some cells look free from nanoparticles – how much cells did uptake the Au-GSH NPs and, if not 100%, how it could affect results of MTT and CAM assays?

Lines 133-137
“The viable canine osteoblasts (CnOB)… … were not included in the calculations.”
- as normal canine osteoblasts served as negative control for extravasation process, authors should include this group in Figure 5 and corresponding calculation.

Lines 165-168
“Some cells injected into the CAM, treated before with Au-GSH NPs, showed intracellular accumulation of Au-GSH NPs (marked with red arrows), visible at t=24h in the CAM (Suppl. 167 Mat. 3)”
- this data should be moved to main manuscript, as it indicates consistency of nanoparticles action between cell culture and CAM. Also, from the figure in supplementary 3 with figure 4 it seems that presented cells are mostly intravascular. Did authors count the cells with visible Au-GSH NPs accumulation? It would be better to present specific data instead of ‘some’. Also, data on ratio of cells with proved accumulation of nanoparticles between cell undergone extravasation and cells that did not would be a good addition.

Lines 177-178
“2.4 Higher MMP-2 expression in canine OS metastasis microenvironment after Au-GSH NPs treatment”
- the section title should be corrected as Figure 8 shows not higher but multidirectional changes depending on cell line.

Line 250

“…proximity to bigger blood vessels of the CAM (Fig. 5BCD)”
- authors refer to Figure 5 but data is presented in Figure 6.

Discussion:
Discussion generally is too much saturated with specific details about previous studies not directly related to current manuscript. Also some parts belong to Materials and Methods section  -

Lines 229-236
“For each treatment group N=10-12 chick embryos at EDD 11 were used for performing the injections. When assessing the extravasation efficiency of OSCA-8 and OSCA-32 cell lines, every embryo used in the study had 2-4 windows placed, each constituting a single region of interest (ROI) (Fig. 6A). Calculations were performed using each ROI as an individual repetition. To calculate the mean extravasation rate, all the windows (ROIs) placed in all embryos in the group were averaged, that is the mean extravasation efficiency value shows the value for the whole group injected with treated or untreated cells. This is in agreement with the methodology published by Kim et al[16].”

Or to Introduction section –

Lines 220-222
“Moreover, it is characterized by several advantages such as time-efficiency, low cost, and better visualization of cancer cells in comparison to other in vivo models, together with the premature chick embryo being naturally immunodeficient.“

And lines 204-217

“The standard procedure in preclinical studies of novel therapies involves in vitro assays and further in vivo murine studies. Nevertheless, in metastasis research, the models that will simultaneously guarantee a proper environment for complex metastatic processes and easy, fast access are needed. Visualizing cancer cells’ extravasation abilities and the influence of potential anti-extravasation therapies can be provided by different methods. Typically used in vitro models, such as the cancer-endothelium adhesion and invasion assay or Transwell assay, have limitations associated with 2D design of the study, lack of in vivo hemodynamics and a full range of blood ingredients, and interactions between them. In the Transwell assay most of the cells eventually translocate over a 24- to 48-h period of incubation which is altering the proper assessment of metastatic abilities of cell lines, which might significantly differ for in vivo conditions [16,33]. Recent constructs such as microfluidic chips with microvascular networks in 3D extracellular matrix (ECM) are more accurate regarding the 3D design, and the possibility to control vascular patterns, but still lack more complex biological features [33,34].”

On the other hand, the authors should discuss more the possible mechanisms of Au-GSH NPs action on extravasation process and meaning of difference in MMP-2 expression changes.

The statement in lines 236-238 “The blood circulation environment is extremely hostile for cancer cells under the process of metastasis, therefore only a small percentage of cells is able to survive and extravasate [39]” seems not very relevant to used model due to CAM lacking lot of mechanisms mentioned in the reference.

The authors mention MMP-2 frequently together with MMP-9 in the discussion, why then authors choose to explore the changes only in MMP-2 in current study?

Materials and Methods
Lines 339-341
“The complex of glutathione stabilized gold nanoparticles (Au-GSH NPs) (4.3 ± 1.1nm) was synthesized and determined using TEM and dynamic light scattering according to a previously described procedure[47,48] .”
- authors should provide brief description in addition to the references.

Statistics section – authors should present here or in appropriate manuscript sections the number of samples used for analysis (extravasation rates, MMP expression, MTT test). Also, some data is presented as mean±sem and some as mean±sd – it is advisable to all data in manuscript in one style.

Abbreviations should be defined on first appearance, please add definition in text for EDD and TME. Also some abbreviations are redundant - DMSO, PBS, RIPA, VEGFR, ZO-1 are used only once in text, NDS is not found anywhere in the manuscript.

References
Reference 51 is inappropriately inserted in line 51, that clearly refers to different work. This reference is not mentioned anywhere else in the manuscript

Reviewer 2 Report

I read carefully this article which presents a work on the inhibition of extravasation of canine osteosarcoma cells by gold nanoparticles.

This article has a lot of weaknesses, first and foremost the writing is messy. The introduction is not well written, the CAM model is poorly described as well as the discussion.

 In the methods chapter, there is often no indication of the temperature at which the experiment was performed.

There is no legend for the supplementary files.

Experiments inhibiting extravasation of canine bone sarcoma cells are not convincing.

It is true that the model used meets the 3Rs but the mouse model has many advantages such as, being able to perform an orthotopic graft and thus have a tumor in a medium corresponding to that observed in humans, moreover with the mouse model and murine sarcoma transplants, there is no concern about a non-mature immune system that can obviously modify the results of the experiment.

How can the authors explain the high mortality of embryos during their experiments?

Do the authors have some hypotheses to explain that there is no difference in the rate of extravasation between the two OS cancer lines that are different aggressiveness ... even all of these canine carcinomas are not yet known.

Reviewer 3 Report

This study is related with analysis of glutathione stabilized AuNPs on the inhibition of extravasation of canine osteosarcoma cells through the use of ex-ovo chick embryo chorioallantoic membrane (CAM) model. Methodology is represented in very good way. The study is interesting and can be helpful in designing treatement strategy for malignancy prevention. However, some changes are required.

Abstract:

Strict revision of abstract is needed. It only comprises of osteocarcinoma introduction and objective of study. Please write these two in first 2-3 lines. Then mention methodology and results. Finally describe outcome in one line.

Results:

Sub-heading 2.1, 2.2, 2.4 are too long. Modify them in a proper and short form.